# Characterization of Alternative Splicing Events in Porcine Skeletal Muscles with Different Intramuscular Fat Contents

**DOI:** 10.3390/biom12020154

**Published:** 2022-01-18

**Authors:** Wanjun Hao, Zewei Yang, Yuanlu Sun, Jiaxin Li, Dongjie Zhang, Di Liu, Xiuqin Yang

**Affiliations:** 1College of Animal Science and Technology, Northeast Agricultural University, Harbin 150030, China; haowanjun1109@163.com (W.H.); yangzewei1997@163.com (Z.Y.); sunyuanlu2021@163.com (Y.S.); ljx3481@163.com (J.L.); 2Institute of Animal Husbandry, Heilongjiang Academy of Agricultural Sciences, Harbin 150086, China; djzhang8109@163.com

**Keywords:** meat quality, intramuscular fat content, alternative splicing, differentially expressed genes

## Abstract

Meat quality is one of the most important economic traits in pig breeding and production. Intramuscular fat (IMF) is a major factor that improves meat quality. To better understand the alternative splicing (AS) events underlying meat quality, long-read isoform sequencing (Iso-seq) was used to identify differential (D)AS events between the longissimus thoracis (LT) and semitendinosus (ST), which differ in IMF content, together with short-read RNA-seq. Through Iso-seq analysis, we identified a total of 56,789 novel transcripts covering protein-coding genes, lncRNA, and fusion transcripts that were not previously annotated in pigs. We also identified 456,965 AS events, among which 3930 were DAS events, corresponding to 2364 unique genes. Through integrative analysis of Iso-seq and RNA-seq, we identified 1174 differentially expressed genes (DEGs), among which 122 were DAS genes, i.e., DE-DAS genes. There are 12 overlapped pathways between the top 20 DEGs and DE-DAS genes, as revealed by KEGG (Kyoto Encyclopedia of Genes and Genomes) analysis, indicating that DE-DAS genes play important roles in the differential phenotype of LT and ST. Further analysis showed that upregulated DE-DAS genes are more important than downregulated ones in IMF deposition. Fatty acid degradation and the PPAR (peroxisome proliferator-activated receptor) signaling pathway were found to be the most important pathways regulating the differential fat deposition of the two muscles. The results update the existing porcine genome annotations and provide data for the in-depth exploration of the mechanisms underlying meat quality and IMF deposition.

## 1. Introduction

Meat quality is emerging as one of the most important economic traits in pig breeding and production, owing to the demands of consumers. Meat quality is a comprehensive indicator composed of the intramuscular fat (IMF) content, pH, tenderness, meat color, water-holding capacity, muscle fiber composition, etc. Most traits of meat quality exhibit low to moderate heritability [1,2], and they are expensive to measure. Selecting a meat quality using traditional methods, such as best linear unbiased prediction, is difficult. As such, molecular biology techniques have been developed to improve meat quality through molecular breeding strategies, for example, molecular marker-assisted selection. Many studies have focused on revealing the mechanisms underlying meat quality, which are the prerequisite for the molecular selection of the trait.

Now, high-throughput characterization of genetic mechanisms underlying meat quality can be performed efficiently with the development of next-generation sequencing technology. Many efforts have been directed at mRNA, LncRNA, miRNA, quantitative trait loci, and single-nucleotide polymorphisms, as well at the genome-wide level, with the aim of elucidating genetic factors affecting meat quality [3,4,5,6], which have revealed key genes and pathways implicated in meat quality. However, no efforts have been devoted to determining the effect of alternative splicing (AS) on meat quality.

AS is a ubiquitous phenomenon in mammals that leads to production of multiple mRNA species from a single gene [7,8]. AS can be grouped into five basic patterns: exon skipping (ES), mutually exclusive exons (MEE), alternative acceptor site (AAS), alternative donor site (ADS), and intron retention (IR) [9,10,11]. It is now well-established that, regardless of whether novel functional protein isoforms are produced, AS is an important mechanism regulating gene expression. It not only increases transcriptome and proteome diversity, but modulates the major mRNA levels by producing multiple mRNAs from the same pre-mRNA. Additionally, AS can regulate the role of other RNAs by competing with them for the same regulators [12].

Through genome-wide analysis, it has been estimated that 90–95% of multi-exon genes undergo AS [10,13]. At present, the human coding genes are annotated with four unique transcripts, on average, in the reference set [14], whereas studies suggested that, on average, at least 10 AS transcripts exist for each human coding gene [15,16]. Many AS transcripts remain to be identified. It was shown that about two-thirds of AS events are tissue-specific in humans [13], and the major isoform varies according to tissue and cell lines [17], indicating the importance of AS in tissue differentiation. AS events have been widely implicated in physiological processes, including tissue identity acquisition, organ development, etc. [18]. Dysregulation of AS can be related to a vast repertoire of diseases [19].

Long-read isoform sequencing (Iso-seq) technology, developed by Pacific Biosciences (PacBio), is capable of precisely and conveniently characterizing AS events through directly sequencing full-length transcripts [20,21,22]. Iso-seq analyses have been performed in humans, chicken, and pigs [23,24,25], in which numerous AS events were newly identified, and the regulation of AS was shown to be rigorous. However, the physiological relevance of these AS events is largely unknown. Additionally, reports on the AS with Iso-seq analysis in pigs were mainly performed on mixed tissues [23,26]; no separate analysis was performed of the skeletal muscles. The role of AS in IMF content remains to be identified.

Longissimus thoracis (LT) and semitendinosus (ST), known to differ in biochemical and histological composition, including connective tissue content, proportions of muscle fiber types, etc., are widely used in studies of meat quality [27,28,29]. They differ in many indicators of meat quality. Compared to ST, LT is more tender, lower in moisture content, and higher in lightness value [30]. The IMF content is also higher in LT than in ST, and lipid deposition occurs earlier and is higher in the intramuscular preadipocytes of LT than those of ST in pigs [27,28,31]. The IMF content is positively correlated with tenderness, juiciness, and flavor [32], and, thus, is important for improving meat quality. Although differences in other factors such as muscle fiber composition can affect pork quality, it is interesting to focus on the different IMF contents between LT and ST and to reveal its underlying mechanisms.

Here, we profiled AS events and identified genome-wide novel transcripts in porcine skeletal muscles using the Iso-seq technique. We characterized the differential (D)AS between LT and ST by combined Iso-seq and short-read RNA sequencing (RNA-seq) analyses, and we highlighted the functional pathways involved in IMF deposition. Our results provide valuable data for further revealing the mechanisms underlying meat quality.

## 2. Materials and Methods

### 2.1. Animals, Tissues, and RNA

LT and ST muscles were sampled from three 210-day-old Min pigs, a Chinese indigenous pig breed. The pigs were provided by the Institute of Animal Husbandry, Heilongjiang Academy of Agricultural Sciences, Harbin, China. The pigs were raised under the same conditions including food, housing, etc. Our animal treatment protocol was approved by the Animal Care Committee of Northeast Agricultural University. LT and ST muscles were collected immediately after slaughter and frozen in liquid nitrogen. Total RNA was isolated with TRIzol reagent (Invitrogen, CA, USA) from each tissue and assessed with a NanoPhotometer^®^ spectrophotometer (IMPLEN, CA, USA) and an Agilent 2100 Bioanalyzer (Agilent Technologies, CA, USA). RNA concentration was measured with a Qubit 3.0 Fluorometer (Life Technologies, CA, USA).

### 2.2. PacBio Library Construction and Sequencing

The PacBio library was constructed with equally mixed RNA from LT and ST muscles (n = 3). The full-length cDNA was synthesized with a SMARTerTM PCR cDNA Synthesis Kit (Clonetech, CA, USA) and Oligo dT primer from 1 µg RNA. PCR amplification was carried out with a KAPA HiFi PCR Kit (Roche, Shanghai, China), and primers were added into the end of full-length cDNA in the reverse transcription. The resultant products were size-selected with AMPure PB magnetic beads (PacBio, CA, USA) to obtain fragments not less than 1 kb. End repair was performed for full-length cDNA, and then the single molecular real-time sequencing (SMRT) dumbbell connector was connected. After digesting the unconnected fragments with exonuclease, the cDNA was purified with PB magnetic beads (PacBio) to obtain the sequencing library. The quality of the Iso-seq library was controlled with Qubit 3.0 (Life Technologies) for accurate quantification and an Agilent 2100 (Agilent Technologies) for size detection. The library was sequenced on the sequel II platform (PacBio) by Frasergen Inc. (Wuhan, China).

### 2.3. Raw Read Processing

The SMRTlink program (http://www.pacb.com/, accessed on 1 December 2021) was used for processing the raw reads to obtain full-length nonconcatemer (FLNC) reads with 15,000 ≥ length ≥ 50, passes number ≥ 3, and predicted accuracy ≥ 99%. Briefly, the raw polymerase reads generated by PacBio sequencing were first filtered and trimmed to produce subreads, and then error correction was carried out to obtain circular consensus sequences (CCSs). The nonconcatemer reads containing 5′ and 3′ adapters were regarded as FLNC reads, amongst which those having a poly(A) tail, FLNC_poly(A), were used for further analysis. Iterative clustering for error correction was applied to obtain consensus sequences. Errors in FLNC reads were further corrected with the high-quality Illumina short reads using LoRDEC with the options of -k 21 -s 3 [33]. The FLNC reads before and after error correction were mapped to the pig reference genome assembly (Sus Scrofa 11.1) using GMAP [34] with parameters of -no-chimeras -n 10, and those reads with a high percent of identity (PID) were used for annotation of loci and isoforms.

### 2.4. Loci and Isoform Annotation

Two sequences that had an overlap of ≥20% at the origin of alignment, at least one exon with ≥20% overlap, and the same transcriptional direction were characterized as the same loci transcript. Isoforms were identified in the same loci through removal of redundant and false-positive gene structure with the following criteria: (i) isoform with inner splice sites identical to the longer one; (ii) those missing the 5′ end; (iii) structures aligned by only one FLNC. In this situation, if all the junctions were annotated in the reference genome or supported by RNA-seq data, the isoforms were retained.

### 2.5. Novel Gene Identification

The annotated results were analyzed to determine the novel isoforms of annotated or novel genes through comparison with data in the reference genomes. The isoforms of novel genes were regarded as those that had an overlap of less than 20% or an overlap of more than 20% but transcribed in a different direction than the annotated genes. The novel isoforms of annotated genes were those having novel splice sites compared to the annotated ones. Additionally, if one of the two aligned sequences was a single exon and the other was not, the FLNC was also regarded as a novel isoform of the annotated gene.

The protein-coding potential was analyzed through screening novel isoforms against the NR, GO, KO, KOG, and Swiss-Prot databases with the BLASTX program using an e-value ≤ 1 × 10^−5^; the remaining ones were analyzed with the CPAT program (version 1.2.2) [35] to identify lncRNA with default parameters.

Fusion transcripts were identified with the following criteria: (i) the FLNC mapped to different annotated gene loci; (ii) more than 20% overlap at each locus; (iii) the junction site was supported by Illumina reads.

### 2.6. Alternative Splicing Analysis

AS events in Iso-seq data were first classified using Astalavista with default parameters [36]. Then, Illumina reads were aligned with Iso-seq data to compare the AS events between LT and ST muscles with rMATS software, and DAS events were identified using junction counts and reads on target. Each of the basic AS events, ES, IR, MEE, AD and AE, includes two isoforms, of which one is longer; thus, the variable sequences could be identified as an included/excluded exon. The expression ratio of exon inclusion (EI) events to both events was applied to evaluate the AS level, and the difference (ΔEI) between the two muscles was calculated to determine the changes in the splicing of all five AS events. The DAS events were characterized based on an absolute ΔEI > 5% and FDR < 0.05.

### 2.7. RNA Sequencing and Data Processing

RNA-seq was performed on the Illumina HiSeq 4000 PE150 platform at Frasergen Inc. (Wuhan) to quantify gene/isoform expression in muscles. A total of 6 libraries were constructed with RNAs from LT and ST of three individuals. The samples were the same as those for Iso-seq. We used 3 µg RNA for each library construction. After removing ribosomal RNA with a Ribo-zeroTM rRNA Removal Kit (Epicentre, Wisconsin, USA) and purifying with ethanol precipitation, libraries were constructed with an NEBNext^®^ UltraTM Directional RNA Library Prep Kit for Illumina^®^ (NEB, Ipswich, MA, USA) according to the manufacturer’s recommendations.

Raw reads were cleaned using SOAPnuke [37]. The high-quality clean reads were mapped to the porcine reference genome (Sus scrofa 11.1) using HISAT2 [38]. Reads mapped perfectly or with one mismatch were employed to assemble transcripts with StringTie under the default parameters [16]. The resultant transcripts were annotated with the gffcompare program. The expression levels were measured using the fragments per kilobase per million bases (FPKM) method using the PacBio GTF annotation file. The gene transcription was compared between two muscles using DESeq2, and those having an absolute log2 fold-change > 1 and FDR < 0.05 were identified as differentially expressed genes (DEGs).

### 2.8. Reverse-Transcription PCR and Real-Time Quantitative PCR

Total RNA was isolated as described above and treated with DNase I to eliminate genomic DNA contamination. Reverse transcription (RT) was performed with the PrimeScriptTM RT Reagent Kit (TaKaRa, Dalian, China). PCR and real-time quantitative PCR (qPCR) was carried out as described previously [39]. RT-PCR products were visualized on 2% agarose gels or 6.5% polyacrylamide gel electrophoresis. Primer information was shown in Appendix A.

## 3. Results

### 3.1. Overview of PacBio Iso-Seq Data

In total, we detected 555,753 polymerase reads representing more than 53 G bases, with an average length of >9 kb. After processing, 36,499,249 filtered subread reads with an average length of 1383 bp and 395,182 circular consensus sequences (CCSs) with an average length of 1769 bp were obtained. CCSs were further classified into non-FL (88,282), FL (306,900), FLNC (293,556), and FLNC_polyA (293,459) reads based on the presence of 5′ adaptor, 3′ adaptor, and poly(A) tails (Figure 1A). FLNC_polyA reads have an average length of 1496 bp (Figure 1B). The data were then corrected with the high-quality short reads of RNA-seq, and a total of 281,522 high-quality FLNC_polyA reads (95.93%) were obtained for further analysis.

After filtering redundancies and false positives, the high-quality FLNC_polyA reads covered 59,075 transcripts (Appendix A) and were aligned to 31,243 loci, of which 9623 were annotated in a reference genome (Figure 1C). Based on the mapping results, the transcripts were classified into three groups: known (2286) and novel transcripts (34,605) of annotated loci and novel transcripts (22,184) that were mapped to unannotated loci (Figure 1A). Of 59,075 transcripts, 34,188 (57.87%) were multi-exon involved in 9518 loci and 24,887 (42.13%) were single exon.

Among the multi-exon genes, 5502 loci had more than one transcript, producing a total of 29,803 transcripts, accounting for 50.45% of the total transcripts obtained. The relationships between the numbers of transcripts and exon or length were analyzed, and we found that genes having more transcripts contain more exons and are longer amongst the multi-exon genes (Figure 1D,E). Full-length frequency analysis showed that 82.93% (170,454/205,544) of annotated multi-exon FLNC_polyA reads had the same initial splice donor sites as the mRNAs in the reference database, indicating the high integrity of our Iso-seq data in the structure. The degree of transcript integrity is similar to previous Iso-seq data in humans and pigs [23,40].

### 3.2. Identification of Novel Gene

A total of 56,789 novel transcripts (34,605 + 22,184) were discovered in this study by Iso-seq. Among the 22,184 novel isoforms of unannotated loci, 20,427 were single exon (Appendix A), accounting for 94.46%, which is far higher than the whole level of Iso-seq data (42.13%). We found that 268 unannotated loci had more than one transcript covering 827 isoforms. Studies have shown that protein-coding genes are usually multi-exon and overwhelmingly alternatively spliced [10,20]. The results suggested that the proportion of protein-coding transcripts in the 22,184 novel isoforms was low.

The novel genes (22,184) were screened against public databases, and the results showed that 11,987 (54.03%), 1168 (5.27%), 1406 (6.34%), 478 (2.15%), and 4254 (19.18%) of them could be found in the NR, GO, KO, KOG, and Swiss-Prot databases, respectively. A total of 266 novel isoforms had significant hits in the five databases (Figure 2A). A total of 10,166 (45.83%) isoforms were not found in any of the protein databases. After filtering out potential lncRNAs amongst them, TransDecoder (http://transdecoder.sf.net, accessed on 1 December 2021) was used to predict coding sequences for the remaining isoforms, of which 80 were identified as novel protein-coding genes (Appendix A). These 80 novel transcripts might be pig-specific or low-conserved amongst species.

A total of 11,371 lncRNAs were identified (Appendix A). The distribution of lncRNAs in the genome is shown in Figure 2B. We also found 19 fusion transcripts that contain sequences from different annotated gene loci (Appendix A). Of the fusion genes, 18 occur interchromosomally and the remaining one is produced by an intrachromosome event, indicating that the interchromosome event was the main source of the fusion gene (Figure 2B). RT-PCR was used to validate the novel genes identified. A total of five isoforms of unannotated loci, four novel isoforms of annotated loci, three lncRNAs, and three fusion transcripts were confirmed (Figure 2C), demonstrating that these are bona fide isoforms/loci.

### 3.3. Alternative Splicing Events

A total of 456,965 AS events were obtained by our Iso-seq analysis, and all of the five basic AS types were covered (Figure 3A). Among the basic AS types, IR and ES were predominant, with IR being the most abundant and MEE being the least, which is consistent with previous studies in animals [23,25]. Moreover, a large number of AS events, such as those containing multiple basic AS patterns, were grouped into other types of AS, indicating the complexity of AS. The AS events of six genes, including one lncRNA and five protein-coding genes, were validated with RT-PCR (Figure 3B), demonstrating the accuracy and potential of Iso-seq in detecting AS events. Among the multi-exon genes, the gene number gradually decreased with transcript number per increasing gene. Overall, the number of multi-exon genes detected with Iso-seq analysis was lower than that in the reference genome; the reason might be that only one differentiated tissue, muscle, was used in the Iso-seq. However, the number of genes with ≥10 transcripts was much higher than that provided by the reference genome (Figure 3C).

Many genes were found to have more transcripts than deposited in the genome database, indicating Iso-seq is an effective tool for the characterization of AS variants. For example, the creatine kinase M-type (CKM) gene (ENSSSCG00000036132) has three transcripts in reference annotation, but 743 isoforms were found in Iso-seq data. Additionally, a protein-coding gene annotated as novel without a name in the ensemble database (ENSSSCG00000010190) was found to have 1095 transcripts by Iso-seq, although only three transcripts were annotated. Li et al. [23] found the largest number of transcripts, ~337, existing in the Sus scrofa actin alpha 1, skeletal muscle (ACTA1) gene through Iso-seq of a pooled set of 38 tissues of Yorkshire pigs in which the skeletal muscle content was trace. Min pigs and the single tissue of skeletal muscle were used here. Thus, AS is breed- and tissue-specific; more isoforms remain to be identified in the pig genome.

### 3.4. Differential Alternative Splicing Events

Through analyzing the ΔEI, a total of 3930 DAS events were identified, among which 2016 have increased EI and 1914 have decreased EI in ST compared to LT (Appendix A). ES and MEE are the top two abundant DAS types, accounting for over 80%, indicating their important role in the differentiation and development of the two muscle tissues, whereas IR events, which are the predominant AS type in the pooled tissues of LT and ST, have much less DAS between these two tissues. The distribution of increased and decreased EI events is balanced in each kind of AS event (Figure 4A). Two DAS events were selected and validated with RT-PCR (Figure 4B).

After integrating the DAS events belonging to the same gene, 3930 DAS events were found to correspond to 2364 unique genes (Appendix A). In this study, transcripts with an expression level of FPKM ≥ 0.1 in at least one of the six samples were defined as expressed as reported in previous studies [41,42]. The expression level of DAS genes is much more abundant compared to that of the total genes identified (Figure 4C, Appendix A). Most of the DAS genes have similar expression levels between the two muscles (Figure 4D).

The DAS genes were annotated into various GO terms of all three functional categories, including cellular component (CC), biological process (BP), and molecular function (MF). In the BP category, biological regulation and metabolic process are among the most highly enriched GO terms with over 1000 genes (Figure 4E, Appendix A). KEGG classification showed that DAS genes are involved in various pathways, and the results were provided on a level 2 hierarchy (Figure 4F, Appendix A). A total of 37 genes are involved in lipid metabolism. On the level 3 hierarchy (KEGG pathway), various pathways are related to fat formation. For example, in the signal transduction catalog, 15 pathways are involved in adipogenesis, including the Hedgehog, FoxO, Wnt, Apelin, Hippo, Jak-STAT, MAPK, PI3K-Akt, cAMP, cGMP-CKG, TGF-β, mTOR, Notch, and Phospholipase D signaling pathways. These pathways include 202 unique genes, accounting for 76.81% of the catalog. In the endocrine system catalog, there are the PPAR (peroxisome proliferator-activated receptor) signaling pathway, adipocytokine signaling pathway, and regulation of lipolysis in adipocyte. Additionally, in the carbohydrate metabolism catalog, some pathways are involved in short-chain fatty acid metabolism, such as butanoate and propanoate.

### 3.5. Integrated Analysis of Differential Alternative Splicing Events and Differentially Expressed Genes

To further evaluate the role of DAS genes in the differential phenotype of LT and ST, the genome-wide mRNA expression profile was compared between the two muscles through integrated analysis of Iso-seq and RNA-seq data. A total of 1174 DEGs (683 upregulated and 491 downregulated) were identified in ST compared to LT (Figure 5A, Appendix A). Among the DEGs, 122 are DAS genes, i.e., DE-DAS genes, of which 70 are upregulated and 52 are downregulated in ST compared to LT (Figure 5B,C, Appendix A). Seven DE-DAS genes were validated with real-time quantitative PCR (Figure 5D). Functional enrichment analysis of DE-DAS genes suggested that they are enriched in 23 BP terms, 14 MF terms, and 9 CC terms (Appendix A). The most highly enriched BP terms, including biological regulation, signaling, and regulation of biological process, are shown in Figure 5E.

DEGs are involved in various KEGG pathways, with many being fat-related: among the top 20 most significant enrichments, 6 are related to fat formation (Figure 6A). DE-DAS genes are also mainly involved in fat-related pathways: 6 of the top 20 significantly enriched pathways are associated with fat deposition (Figure 6B). Although only 122 out of 1174 DEGs are differentially spliced, there are various KEGG pathways significantly enriched by both DEGs and DE-DAS genes; among the top 20 pathways, 12 are shared, with 5 being fat-related, indicating DAS of DEGs plays major roles in the differential phenotype of the two muscles.

Further analysis showed that upregulated DE-DAS genes are involved in more pathways related to fat formation. Among the top 20 significant enrichments, 7 are fat-related. Four out of the top five pathways are fat-related (Figure 6C). Only two fat-related pathways are significantly enriched by the downregulated DE-DAS genes (Figure 6D). Additionally, upregulated DE-DAS genes share five fat-related pathways with all the DE-DAS genes, whereas no fat-related pathways overlap between downregulated and all DE-DAS genes. These indicated that upregulated DE-DAS genes are the key factors leading to the difference in IMF content between LT and ST. The fatty acid degradation and PPAR signaling pathway are the most important pathways regulating the differential fat deposition of the two muscles.

### 3.6. Transcription Factors in DE-DAS Genes

Among the top 20 up- and downregulated DEGs, 18 are annotated in NR databases and 8 are transcription factors (TFs) (Table 1); additionally, a total of 111 DEGs were identified as TFs through searching the AnimalTFDB database (http://bioinfo.life.hust.edu.cn/AnimalTFDB/, accessed on 1 December 2021) with hmmscan program (Appendix A). There are 71 upregulated and 40 downregulated TFs in ST compared to in LT. These TFs belong to various families, with the most being from the homeobox family (Figure 7A). Of these differentially expressed TFs (DE-TFs), 16 are DAS genes (DE-DAS-TFs) (Figure 7B,C). PPI analysis was performed to identify the interaction of the DE-DAS-TFs with DEGs, and the results showed that ACTN2 and RNF41 play a key role in regulating DEGs (Figure 7D).

## 4. Discussion

In this study, we used Iso-seq combined with RNA-seq techniques to analyze AS events in the ST and LT muscles of pig and found that 122 DE-DAS genes are key regulators in the differential phenotype of the two muscles. Additionally, a large number of AS events and novel transcripts that were not previously annotated in pigs covering protein-coding genes, lncRNA, and fusion transcripts were obtained using PacBio sequencing. The results contribute to further revealing the mechanisms underlying meat quality and are of value for the refinement of the porcine genome annotation as well.

This study was designed to characterize AS in muscles, which is different from previous studies aiming at maximizing transcript diversity with multiple tissues via the SMRT methodology in pigs [25]. Thus, the number of transcripts obtained here is relatively low, but, in agreement with previous studies, our results further emphasize the complexity of the porcine transcriptome and the universality of AS in animals. The percent of AS in multi-exon genes is 57.81% (5502/9518) in our Iso-seq data, much lower than the values reported in previous studies (>90%) [10,13,25]. This finding might also be explained by only one tissue, skeletal muscle, being used here.

A total of 34,605 novel isoforms, previously unannotated, were identified with Iso-seq in 9623 known loci, with an average number of 3.6 per locus, which is a notable increase in the identified transcripts in the porcine genome. Moreover, a huge number of unannotated loci was found. These sequences were validated with short-read data of RNA-seq, homologous sequences in other species, and RT-PCR. Tissue-specific splicing and the subsequent transcripts exist extensively in animals. In a study involving nine porcine tissues, 44% of all detected transcripts were tissue-specific [26]. At the protein level, evidence for tissue-specific splicing was also reported by Rodriguez et al. [43] in multiple tissue groups, including nervous, muscle, blood, digestive, urinary, liver, reproductive, respiratory, endocrine, and placenta tissues. Thus, the many novel transcripts obtained here might be muscle-specific and related to the development, differentiation, and function of muscles. Additionally, Min pigs are an indigenous Chinese pig breed with higher IMF content than that in western pig breeds and were first used here for Iso-seq analysis. There should be a large number of transcripts specific for Min pigs, which might be valuable for revealing the difference in meat quality between Min and other pig breeds. Moreover, the low level of expression and/or conservation among species (such as lncRNA) might be another reason why they were not identified before. Nevertheless, these newly identified sequences not only update the pig transcriptome, but offer clues for the identification of skeletal muscle-specific isoforms, which will contribute to revealing mechanisms underlying skeletal muscle development.

Iso-seq technology can produce full-length transcripts without the aid of assembly, thus providing superior evidence for differentiating AS events. IR events were identified as the most abundant in the mixed samples of LT and ST muscles with Iso-seq. IR was thought to be the most prevalent AS type in various studies [25,44,45]. However, there were different results. For example, AD was the most common form with 44% of the total AS events analyzed, including ES, AD, AA, and IR, while IR accounted for approximately 19% in 34 different tissues, including muscle from Duroc pigs [42]. Additionally, AA was identified as the most prevalent AS type in muscle from a single White cross-bred pig [26]. These results showed that AS is common, complicated, and highly regulated, suggesting its importance in physiological processes. Nevertheless, IR has been emerging as a mechanism underlying gene regulation. It has been found to be strongly regulated and involved in early chick embryo development [25], granulocyte differentiation [46], terminal erythropoiesis [47], and germ cell differentiation [48]. Recent genome-wide splicing analyses showed that increases in IR are age-associated and conserved across species [49,50,51]. The prevalence of IR in muscle of Min pigs implicates its involvement in skeletal muscle development.

However, ES and MEE were characterized as the top two events that differentially splice between two muscles using integrated Iso-seq and RNA-seq analysis, indicating ES and MEE are critical splicing regulatory mechanisms for the different LT and ST phenotypes. ES and MEE are likely to cause amino acid deletion/insertion (indel) in the polypeptide without frameshift or premature termination codon (PTC), which are often present in IR events and result in truncated protein [25]. PTC-containing transcripts might be degraded by the nonsense-mediated mRNA decay mechanism [52,53]; thereafter, IR events function mainly in regulating the levels of productive mRNA and the corresponding protein. The novel isoforms produced by amino acid indels might be a major reason for the differences between LT and ST.

AS plays critical roles in phenotype determination, tissue differentiation, and development [12]. DEG analyses have been performed in porcine skeletal muscle previously, and key genes and pathways related to skeletal muscle development and growth [54,55] and meat quality [56,57] were identified. However, RNA splicing in skeletal muscle was not deeply investigated. Here, mechanisms underlying the differential phenotype of LT and ST were analyzed on the AS level, and pathways involved in IMF deposition were highlighted. Through integrated analysis of Iso-seq and RNA-seq, DAS and DE-DAS genes were identified and were found to be involved in various fat-related pathways, indicating AS plays an important role in the different IMF contents of the two muscles. Although much lower than DEGs in number (122 DE-DAS genes vs. 1177 DEGs), DE-DAS genes share the most pathways (12 out of the top 20) with DEGs and play similar roles in fat deposition, as revealed by KEGG analysis, which suggested that the DAS of DEGs is the major factor affecting the IMF contents in LT and ST muscles. These results indicated that DE-DAS genes are critical for the differential phenotype of ST and LT muscles, especially in fat deposition. The AS of these genes, especially upregulated ones, should be considered first in future studies on meat quality.

## 5. Conclusions

In conclusion, we obtained a full-length pig transcriptome using PacBio Iso-seq. Numerous novel transcripts covering protein-coding genes, lncRNA, and fusion transcripts that were not previously annotated in pigs were identified. A total of 456,965 AS events, of which 3930 are DAS, corresponding to 2364 unique genes, were obtained in LT and ST muscles. The results update the existing genome annotations and revealed the breed and tissue specificity of AS and isoforms. Furthermore, DAS of DEGs were identified as the important factors influencing the IMF contents in LT and ST muscles, and fatty acid degradation and PPAR signaling pathway were found to be the most important pathways regulating the differential deposition of two muscles. The findings provide a valuable basis for the in-depth exploration of the mechanisms underlying meat quality and the IMF deposition.

## Figures and Tables

**Figure 1 biomolecules-12-00154-f001:**
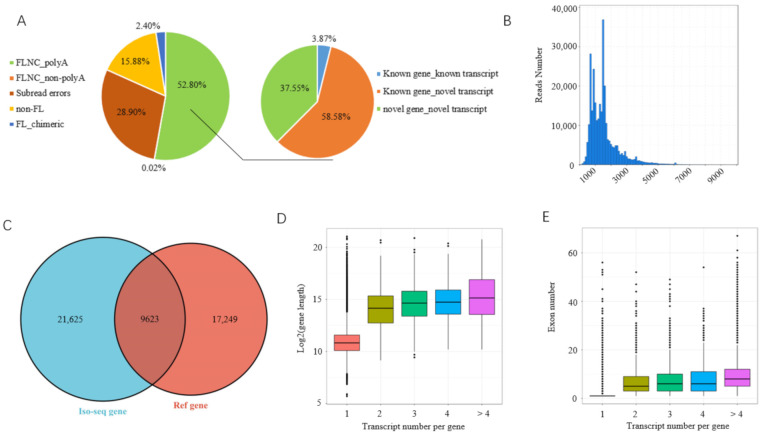
Analysis of Iso-seq data. (**A**) Classification of raw Iso-seq reads; (**B**) length distribution of full-length nonconcatemer reads; (**C**) Venn diagram depicting overlapped genes between Iso-seq and reference genome data; (**D**) analysis of gene length amongst genes generating different numbers of transcripts; (**E**) analysis of exon numbers among genes generating different numbers of transcripts.

**Figure 2 biomolecules-12-00154-f002:**
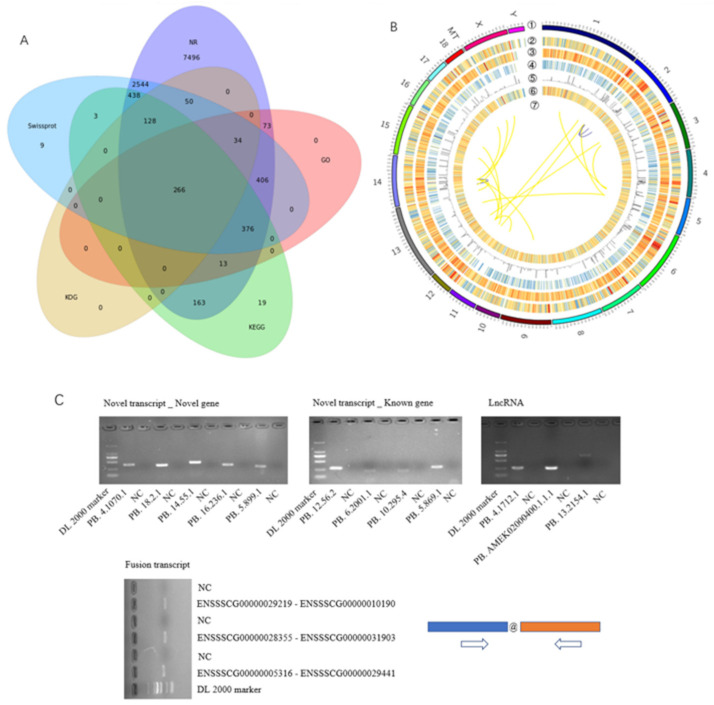
Identification of novel genes by Iso-seq analysis. (**A**) Venn diagram depicting novel protein-coding isoforms among NR, GO, KO, KOG, and Swiss-Prot databases; (**B**) CIRCOS visualization of data identified at the chromosomal level. ① Pig chromosomes; ② distribution of gene in reference genome; ③ distribution of gene identified by Iso-seq; ④ transcript distribution; ⑤ alternative splicing event distribution; ⑥ lncRNA distribution; ⑦ fusion transcripts: inter-chromosome (yellow), intro-chromosome (blue); (**C**) validation of novel isoforms with RT-PCR. Arrow, primer position; @, junction of two genes; NC, negative control.

**Figure 3 biomolecules-12-00154-f003:**
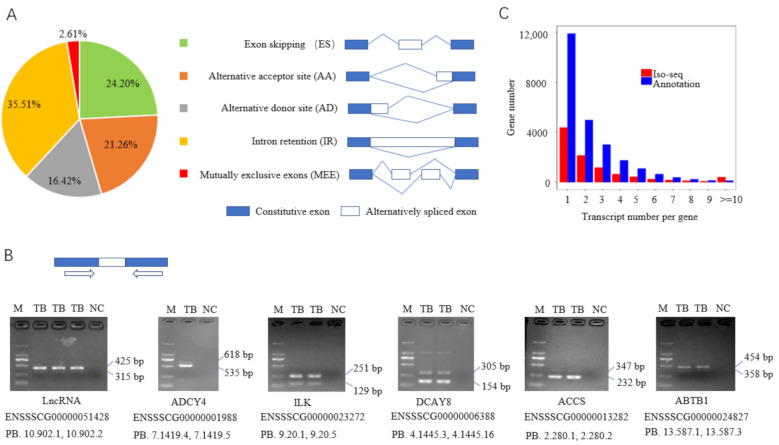
Identification of alternative splicing (AS) events by Iso-seq analysis. (**A**) Distribution of five basic AS events detected, and a schematic illustration of the five AS models; (**B**) RT-PCR validation of alternative splicing events; (**C**) distribution of transcript number per gene among multi-exon genes. M, DL2000 marker; TB, target band; NC, negative control.

**Figure 4 biomolecules-12-00154-f004:**
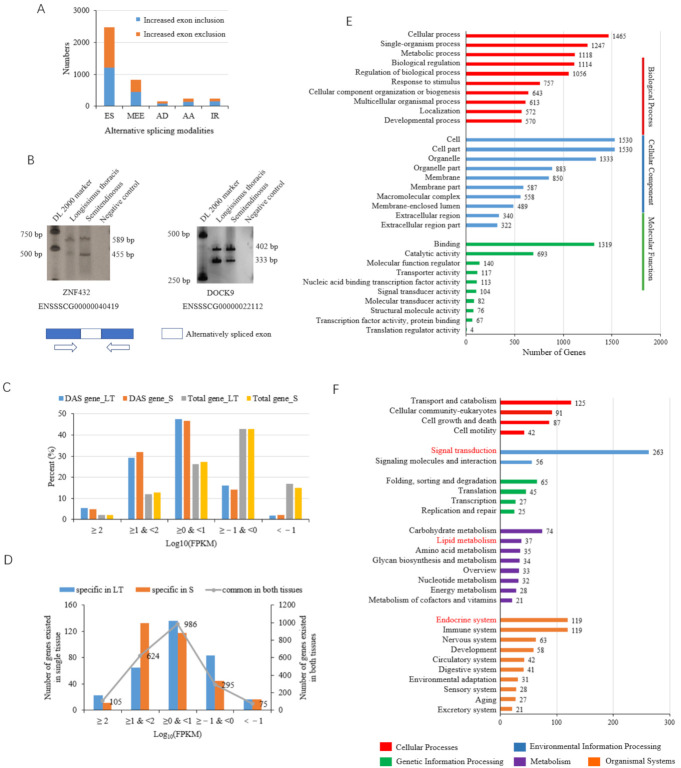
Characterization of differential alternative splicing (DAS) events through integrated analysis of Iso-seq and Illumina-seq. (**A**) Statistics of differential DAS events; (**B**) RT-PCR validation of DAS genes; (**C**) distribution of DAS genes with different expression levels compared to that of total genes identified; (**D**) comparison of expression level of DAS genes between longissimus thoracis and semitendinosus; (**E**) top 10 GO terms enriched by DAS genes in each functional category; (**F**) KEGG (Kyoto Encyclopedia of Genes and Genomes) pathways enriched with over 20 DAS genes.

**Figure 5 biomolecules-12-00154-f005:**
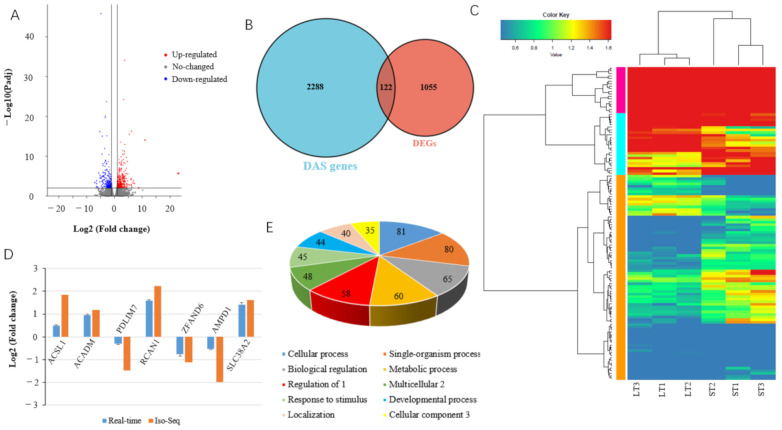
Characterization of differentially expressed (DE) differential alternative splicing (DAS) genes in semitendinosus compared to longissimus thoracis muscle. (**A**) Volcano plot of DE genes; (**B**) Venn diagram depicting DE-DAS and DE genes; (**C**) heatmap of DE-DAS genes; (**D**) real-time PCR validation of DE-DAS genes; (**E**) top 10 biological process categories classified by DE-DAS genes. The gene number is given in the chart. 1, Regulation of biological process; 2, multicellular organismal process; 3, cellular component organization or biogenesis.

**Figure 6 biomolecules-12-00154-f006:**
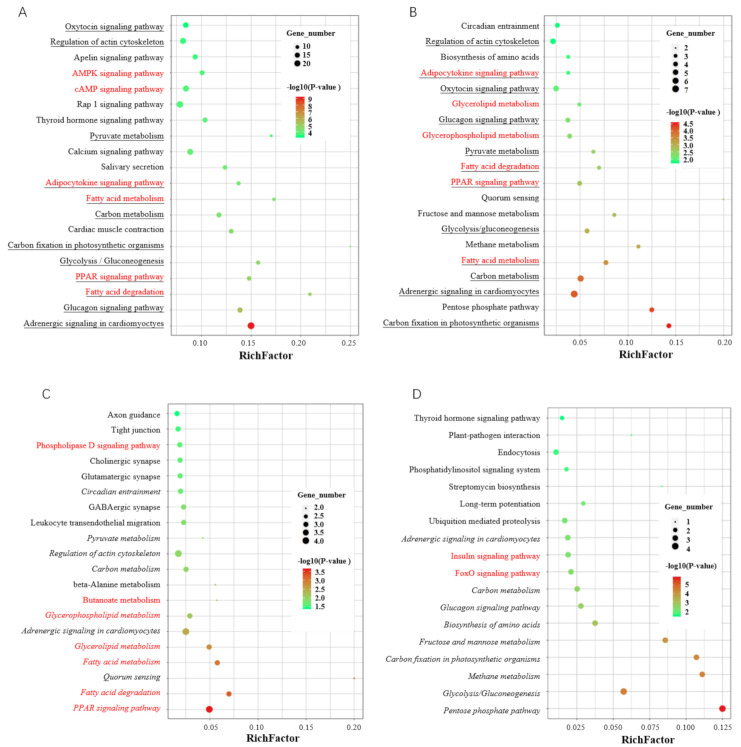
Functional characterization of differentially expressed (DE) differential alternative splicing (DAS) genes. (**A**) Top 20 KEGG pathways significantly enriched by DE genes (DEGs); (**B**) top 20 KEGG pathways significantly enriched by DE-DAS genes; (**C**) top 20 pathways significantly enriched by upregulated DE-DAS genes; (**D**) all pathways significantly enriched by downregulated DE-DAS genes. The pathways involved in fat formation are indicated in red, and those shared by the top 20 of DEGs and DE-DAS are underlined; all pathways overlapped between enrichments of all and upregulated or downregulated DE-DAS genes are indicated in italics.

**Figure 7 biomolecules-12-00154-f007:**
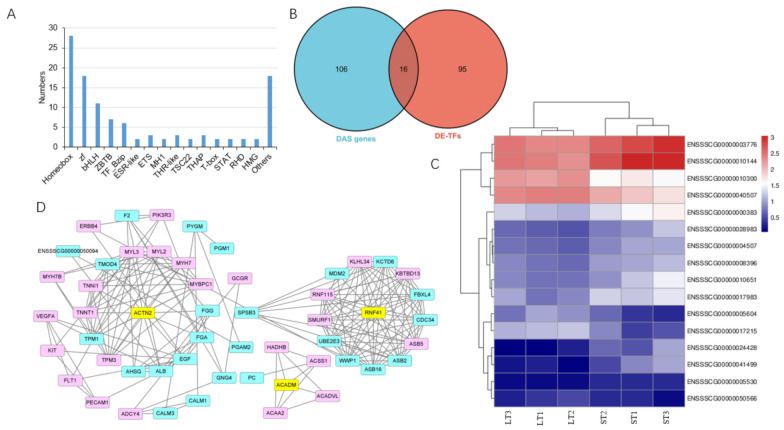
Characterization of differentially expressed transcription factors (DETFs) among differential alternative splicing (DAS) genes. (**A**) family analysis of DETFs; (**B**) Venn diagram of DETFs and DAS genes; (**C**) heatmap of DAS-DETFs; (**D**) protein–protein analysis of DAS-DETFs with differentially expressed genes in semitendinosus compared to longissimus thoracis muscles. Pink, upregulated genes; blue, downregulated genes; yellow, DAS-DETFs.

**Table 1 biomolecules-12-00154-t001:** Top 20 up- and downregulated differentially expressed genes.

Gene ID	Log2FC	NR_Protein_Accession	NR_Defination
7.1204	22.98	*	*
5.156	10.99	XP_013853022.1	^△^^#^ homeobox protein Hox-C9 (Sus scrofa)
ENSSSCG00000036741	10.19	XP_019277494.1	^△^^#^ pituitary homeobox 1 (Panthera pardus)
ENSSSCG00000016698	8.79	XP_003134898.2	^△^^#^ homeobox protein Hox-A11 (Sus scrofa)
ENSSSCG00000029666	8.70	XP_014964404.1	^△^^#^ homeobox protein Hox-A13 isoform X2 (Ovis aries musimon)
18.79	7.16	ABR01162.1	endonuclease/reverse transcriptase (Sus scrofa)
3.205	6.87	*	*
ENSSSCG00000022980	6.62	XP_005669066.1	^△^^#^ T-box transcription factor TBX4 isoform X1 (Sus scrofa)
10.351	6.27	XP_012920268.1	^△^^#^ tigger transposable element-derived protein 1 (Mustela putorius furo)
ENSSSCG00000033532	6.16	XP_003356130.2	^△^ serine/threonine-protein kinase SBK2 (Sus scrofa)
ENSSSCG00000015069	−7.06	NP_001002801.1	apolipoprotein C-III precursor (Sus scrofa)
ENSSSCG00000040910	−6.54	XP_003131314.1	^△^ beta-2-glycoprotein 1 isoform X1 (Sus scrofa)
ENSSSCG00000023686	−6.47	NP_999377.1	transthyretin precursor (Sus scrofa)
ENSSSCG00000011799	−6.44	XP_005652426.1	^△^ alpha-2-HS-glycoprotein (Sus scrofa)
ENSSSCG00000011692	−6.40	XP_003358647.1	^△^^#^ zinc finger protein ZIC 1 (Sus scrofa)
ENSSSCG00000048779	−6.10	XP_013843157.1	^△^^#^ zinc finger protein 646 isoform X1 (Sus scrofa)
ENSSSCG00000032321	−5.86	NP_001001859.1	alpha-1,3-mannosyl-glycoprotein 4-beta-N-acetylglucosaminyltransferase C (Sus scrofa)
ENSSSCG00000005488	−5.83	XP_005660428.1	^△^ alpha-1-acid glycoprotein isoform X1 (Sus scrofa)
ENSSSCG00000042542	−5.78	ABR01162.1	endonuclease/reverse transcriptase (Sus scrofa)
ENSSSCG00000005485	−5.76	NP_001157478.1	protein AMBP precursor (Sus scrofa)

FC, fold change; *, sequence was not identified in the NR database; ^△^, sequence was predicted by automated computational analysis in the NR database; ^#^, sequence is a transcription factor.

## Data Availability

All the relevant data are provided along in the manuscript as Supplementary Files.

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
