# Peer review of "Characterization of Alternative Splicing Events in Porcine Skeletal Muscles with Different Intramuscular Fat Contents"

_biomolecules, 2022, doi:10.3390/biom12020154_

Round 1

Reviewer 1 Report

biomolecules-1516378 - Characterization of alternative splicing events in porcine skeletal muscles with different intramuscular fat content. The manuscript is well written and shows a new approach to gene splicing analysis based on a new solution long-read whole transcriptome sequencing. The manuscript delivers mainly methodological information than biological significance. Because the authors focused on IMF content between LT and ST muscle, these muscles,  besides being skeletal, have significant differences in muscle fibre content and other parameters, which is reflected in transcriptomic profiles. If authors would like to evaluate only gene expression and AS events strongly associated with IMF they should compare the same muscle with different IMF levels. But I think that this manuscript is important due to its metrological side. Because it indicates how the public genetic databases are poor and how much there is to be done in this regard. I have one concern, though I did not understand well, Iso-seq was performed for one library based on six samples obtained from LT and ST muscles, or for two LT and ST? If for two, please describe better in the sub-section Pac bio library construction and sequencing. If, for one, I understand that Pac bio sequencing is costly, how did the authors align AS events with particular muscles? This observation can be done based on RNA-seq results, of course, and AS events can be compared with Iso-seq do such appear at all. It was not exactly clear. Would the authors please explain and add it into material and method section?

Minor comments:

Introduction:

Here should be mentioned that analysed muscles have a little different specify.

Results and discussion:

This section should name results because the discussion is a separate section.

Read length (Figure 1B) after iso-seq looks a little strange; how authors explain significant enrichment of signals  for reads approx. 2000 bp and 1000 bp, when this distribution should present a little different profile  https://dnatech.genomecenter.ucdavis.edu/2017/03/09/pacbio-sequel-version-2-chemistry-and-new-iso-seq-protocol/

Sections of identification of novel gene and alternative splicing events are interesting.

However, in functional analysis, the authors should be more careful and not underline only the lipid metabolic associated process. The differences between these two muscles are broader than only IMF, and the transcriptome profile shows other activities related to muscle fibre well. Nevertheless, it can be mentioned in the introduction section that in this paper the authors focused only on IMF level differences, although these muscles are various in numerous features. If authors would like to evaluate only gene expression and AS events strongly associated with IMF they should compare the same muscle with different IMF levels.

Reference:

The authors should check the reference, there is something confused. For example nr 30 did not consider IMF level in analysed muscles

Reviewer 2 Report

The work is well structured and presented a huge quantity of data. Unfortunately the quality of pictures is really low so the amelioration of the pictures is mandatory before publishing the article. SImilary, the english is not really good and large parts of Introduction need to be rephrased. Regarding the pictures, I would like to see the negative control in PCR gel in order to avoid contaminations that could mislead the results

Indeed, as the authors annotated in Conclusion Section:

“Numerous novel transcripts covering protein-coding genes, lncRNA, and fusion tran-scripts that were not annotated in pigs previously were identified.”

I would like that they could discuss deeper this point, considering the role of these genes in biology of porcine muscles.
